# Symmetry Breaking of Electronic Structure upon the π→π* Excitation in Anthranilic Acid Homodimer

**DOI:** 10.3390/molecules29235562

**Published:** 2024-11-25

**Authors:** Marcin Andrzejak, Joanna Zams, Jakub Goclon, Przemysław Kolek

**Affiliations:** 1K. Gumiński Department of Theoretical Chemistry, Faculty of Chemistry, Jagiellonian University, 31-007 Kraków, Poland; joanna.zams@doctoral.uj.edu.pl; 2Doctoral School of Exact and Natural Sciences, Jagiellonian University, 31-007 Kraków, Poland; 3Faculty of Chemistry, University of Bialystok, 15-328 Białystok, Poland; j.goclon@uwb.edu.pl; 4Institute of Physics, University of Rzeszów, 35-310 Rzeszów, Poland

**Keywords:** carboxylic acid dimers, symmetry breaking, ab initio correlated methods: CC2, PC-NEVPT2, one-electron transition density matrix, excited states, hydrogen bonds

## Abstract

The main purpose of this study is to characterize the nature of the low-energy singlet excited states of the anthranilic acid homodimer (AA_2_) and their changes (symmetry breaking) caused by deformation of the centrosymmetric, ground state structure of AA_2_ towards the geometry of the S_1_ state. We employ both the correlated ab initio methods (approximate Coupled Clusters Singles and Doubles—CC2 and CASSCF/NEVPT2) as well as the DFT/TDDFT calculations with two exchange–correlation functionals, i.e., B3LYP and CAM-B3LYP. The composition of the wavefunctions is investigated using the one-electron transition density matrix and difference density maps. We demonstrate that in the case of AA_2_, small asymmetric distortions of geometry bring about unproportionally large changes in the excited state wavefunctions. We further provide comprehensive characterization of the AA_2_ electronic structure, showing that the excitation is nearly completely localized on one of the monomers, which stands in agreement with the experimental evidence. The excitation increases the π-electronic coupling of the substituents and the aromatic ring, but only in the excited monomer, while the changes in the electronic structure of the unexcited monomer are negligible (after geometry relaxation). The increased electronic density strengthens both intra- and intermolecular hydrogen bonds formed by the carbonyl oxygen atom of the excited monomer, making them significantly stronger than in the ground state. Although the overall pattern of changes remains qualitatively consistent across all methods employed, CC2 predicts more pronounced excitation-induced modifications of the electronic structure compared to the more routinely used TDDFT approach. The most important deficiency of the B3LYP functional in the present context is locating two charge-transfer states at erroneously low energies, in close proximity of the S_1_ and S_2_ states. The range-corrected CAM-B3LYP exchange–correlation functional gives a considerably improved description of the CT states at the price of overshot excitation energies.

## 1. Introduction

Carboxylic acid dimers have been extensively studied because of the presence of two strong intermolecular hydrogen bonds, which enable a variety of intriguing phenomena regarding molecular structure, dynamics, and spectroscopy [1,2,3,4,5,6,7,8,9,10,11], just to mention the possibility of proton transfer or double proton transfer between carboxylic groups [12,13,14,15], as well as excitonic coupling of monomers resulting in Davydov (excitonic) splitting [16,17,18].

The S_0_→S_1_(ππ*) excitation brings about a distinct asymmetry of the molecular properties related to the intermolecular vibrational degrees of freedom. The excitation leads to considerable strengthening of one of the intermolecular hydrogen bonds that link the carboxylic groups of both acid molecules in the dimers (of benzoic acid BA_2_ [1,4,5,6,7,8,9,10,11,12,13,14,15], salicylic acid SA_2_ [19,20,21], and anthranilic acid AA_2_ [22,23]) and to the bent geometry of the excited dimers (symmetry lowering from C_2h_ to C_s_ due to the in-plane distortion). The distortion is most apparent in the excitation spectra, since it generates a strong progression (BA_2_ [6,8,9], SA_2_ [14], and AA_2_ [22,23]) in the so-called geared (cogwheel) inter-monomer mode of frequency 57–60 cm^−1^, associated with the asymmetric mode of B_u_ symmetry in the ground state, which is forbidden in the ground-state symmetry C_2h_. The bent geometry of the dimers in the S_1_ excited state is also validated by the rotational constants fitting to the high-resolution spectra of BA_2_ in a supersonic jet, indicating that the excited-state structure of BA_2_ is bent by 3.4 ± 1.7 degrees [15]. The above suggests at least some degree of localization of the electronic excitation on one of the monomers, with one of the monomers being significantly different than the other.

Furthermore, the dimers of the aromatic acids ortho substituted with hydroxyl group (SA_2_ [19,20,21]) or amino group (in AA_2_ [22,23]) provide the evidence for the practically complete localization of excitation, which is based on the intramolecular degrees of freedom. In these molecules, the lowest ππ* excitation induces a significant excited-state intramolecular dislocation of the hydrogen atom [20,21,24,25,26,27,28,29] along the OH···O or NH···O hydrogen bond and its enormous strengthening. In the dimers, however, it takes place in only one monomer, while the corresponding H-bond in the other monomer remains unaffected. This causes a significant weakening of the covalent bond involving the H atom and a significant decrease in the O–H (in SA_2_) or N–H (in AA_2_) stretching frequency. The latter is directly evident in the IR-UV and UV-IR double resonance spectroscopy (SA [20], AA [28,29]), proving that in the equilibrium geometry of the S_1_ excited state of the dimers, the electronic excitation is localized on one monomer, while the second one remains practically in the ground state. Clearly, the symmetry breaking concerns both the molecular geometry and dynamics (the force constants and the bond energies), causing decoupling of vibrational modes of the two monomers (for BA_2_ and SA_2_ [8,9] and for AA_2_ studied in our recent work [23]).

On the other hand, for benzoic acid dimer BA_2_, the Davydov splitting (vibrationally quenched) of several wavenumbers was observed between vertical excitation energies to the S_1_ and S_2_ adiabatic states [18], owing to a very small symmetry breaking imposed by an asymmetric isotopic substitution. This indicates that for the symmetric ground-state geometry (if fine isotopic effects on molecular geometry are neglected), the S_1_ and S_2_ excited states are almost degenerate and the high symmetry of the ground state is essentially retained.

The localized excitation resulting in the symmetry lowering can be viewed in a broad context of the photoinduced symmetry breaking [30,31,32,33]. This effect is essential for photoinduced symmetry-breaking charge transfer/separation (SB-CT and SB-CS, respectively) and plays an important role as an underlying photochemical mechanism of photosynthesis and for solar energy harvesting. Recently, symmetry-breaking phenomena have been observed in the ground state of carboxylic acid dimers at cryogenic temperatures [34,35].

The properties of AA monomer [28,29,36,37,38] and dimer [22,23] in the S_1_ excited state were thoroughly studied in a supersonic jet using LIF excitation spectra [22,23,28,36,37,38,39], as well as dispersed fluorescence, REMPI-TOF, hole burning, and IR-UV/UV-IR double resonance spectroscopy [22,28,29]. A comprehensive introduction to the issues concerning the excited state of the AA_2_ is included in the articles by Southern et al. [22] and recently by Kolek et al. [23]. The only detected form of the AA moiety both for the AA monomer and dimer is the rotamer, in which the carbonyl oxygen atom forms the intramolecular hydrogen bond with the amino H atom (see Figure 1). This rotamer is more stable than the other one (in which the singly bonded oxygen atom is involved in the NH···O hydrogen bond) owing to the formation of a stronger intramolecular hydrogen bond.

The main goal of this work is to study the initial phases of the process of localization of the S_1_ excitation (on one of the monomers) and the symmetry breaking of the C_2h_ wavefunctions of the dimer, and then to investigate the impact of the final near-complete localization on the electronic characteristics of the lowest excited singlet state of the dimer (electron density, bond orders, the nature of higher energy excitations). In order to increase the credibility of the results, we conducted quantum chemistry calculations using both the high-level ab initio methods (CC2, CASSCF/NEVPT2) combined with the triple-ς basis set (def2-TZVPP [40]) and the somewhat more routine DFT/TDDFT calculations with the ubiquitous B3LYP exchange correlation functional and its range-corrected counterpart—CAM-B3LYP—so that their performance can be assessed against the ab initio methods. High-quality measurements from the supersonic jet expansion are eventually used as references for the S_1_ state adiabatic energies obtained in our calculations.

## 2. Computational Details

Most of the correlated ab initio calculations for the monomer and dimer of AA were carried out using CC2 [41], which is a cost-effective variant of the CCSD model with both geometry optimization and harmonic analysis capabilities [42,43,44], applicable to small and medium-sized molecules. In the CC2 model, the amplitude for double excitations is added perturbatively to the CCS wavefunction, but then the amplitudes for the single excitations are reoptimized in their presence, and the process is repeated until self-consistency is achieved. The CC2 calculations formally scale as N^5^ (times the number of iterations), N being the number of orbitals. Using resolution of identity with auxiliary basis sets and a partitioned form of the CC2 equations, it can be lowered to approximately N^4^, with significantly reduced memory and disk space requirements, which enables handling 2–3 times larger systems than with the conventional approach. The errors in excitation energies caused by the RI approximation are negligible compared to the correlation error inherent in the CC2 method [45]. The excited states are calculated by applying linear response theory to the CC2 ground state wavefunction [46]. Using a Turbomole implementation of CC2, we performed geometry optimizations at the ground state electronic level as well as for the S_1_ electronic state of both the monomer and the dimer of AA. We further computed the energies and properties of several higher-energy excited states of the AA dimer for its ground state structure, the structure of the S_1_ state, and for a series of intermediate geometries to trace the evolution of the excited states with changing geometry. We subsequently conducted analogous calculations at the DFT/TDDFT level of theory using the B3LYP [47,48,49] and CAM-B3LYP [50] exchange–correlation (XC) functionals (including the D3 dispersion correction [51] with Becke-Johnson damping [52] for both geometry optimizations and the single-point calculations of the excited state energies), in order to test their performance against the results of more advanced ab initio calculations.

On top of the CC2 geometries, we computed the energies and wavefunctions for the ground state and four excited singlet states at the state-averaged CASSCF level of theory using the well-balanced active space of eight π-orbitals containing eight electrons (8/8). In order to account for the dynamic electron correlation effects, the CASSCF calculations were followed by the single-state PC-NEVPT2 [53,54] treatment (the loss of orthogonality of the CASSCF states due to the NEVPT2 corrections was tested in the multi-state calculations and was found to be negligible for the studied states of the AA dimer). Apart from the independent verification of the transition energies obtained earlier in the single-reference approaches, the multireference calculations yielded the Mayer bond orders [55,56] for both the S_0_ and S_1_ excited states. All the calculations were obtained using the def2-TZVPP basis set on all atoms. The exploratory calculations with the same basis set furnished with additional diffuse basis functions (def2-TZVPPD [40]) showed that for the electronic states under study, inclusion of these functions had negligible impact on the transition energies; therefore, for the sake of consistency of the theoretical description, we have refrained from further use of the diffuse functions. All the single reference calculations were carried out using the Turbomole 7.8.1 package [57] and the multireference ones that were conducted using the ORCA 6.0 program [58].

The character of the excited states was analyzed in greater detail in terms of the one-electron transition (or difference) density matrices using the TheoDORE 3.2 program by Plasser et al. [59], which allows for quantification of the contributions of the local and charge-transfer excitations between arbitrarily chosen molecular fragments (which for the studied dimer were naturally the AA monomers).

## 3. Results and Discussion

### 3.1. Excited Electronic States of AA_2_

Combinations of the S_1_ (L_a_) states of AA molecules generate the adiabatic S_1_ and S_2_ states of AA_2_, of the vertical excitation energy of ca. 4.0 eV, and the Davydov (excitonic) splitting around 300 cm^−1^ (0.04 eV) depending on the quantum chemistry method (see Table 1). However, at the equilibrium geometry of the S_1_ state, these states are sufficiently separated on the energy scale (by 0.5–0.7 eV, depending on the method used). The adiabatic excitation energies for the S_1_ state of AA and AA_2_ (see Table 1) located by the CC2 and B3LYP/TDDFT methods at 3.63–3.65 eV are after the ZPE correction that lowers the excitation energy by 0.10–0.16 eV (for AA_2_) in excellent agreement with the experimental origin band energy observed in the supersonic jet spectra (3.54 eV for AA [28,37] and 3.50 eV for AA_2_ [22,23]. The NEVPT2 adiabatic energy of the S_1_ state is only slightly overestimated (3.61 eV for AA_2_). This can be put down either on the molecular geometries, which had been optimized at the CC2 level of theory and, as such, were not strictly compatible with the NEVPT2 method, or on the relatively modest active space comprising eight π-orbitals. Any extensions of the active space include more π-orbitals (either occupied or virtual); however, they showed that the additional orbitals had occupations close to 2 or 0. It demonstrates that the four lowest-energy excited states of the dimer are well-described by the configurations constructed from the eight initially selected active orbitals. Also, the active orbitals with occupations close to 0 or 2 lead to inevitable convergence problems of the CASSCF calculations. We therefore refrained from building up larger active spaces, especially since the error of our NEVPT2 result of 0.11 eV was within the 0.13 eV margin, which is the expected accuracy of the PC-NEVPT2 method [60]. Besides the excitation energies, the multireference calculations yielded the Mayer bond orders [55,56] for both the S_0_ and S_1_ excited states. Note that in the CASSCF approach all the states are treated on equal footing (unlike the ground state in most single reference methods), which lends credibility to comparisons made between the S_1_ and S_0_ bond orders obtained in this way.

The CAM-B3LYP/TDDFT approach quite significantly overshoots the adiabatic excitation energy (by over 0.3 eV), which is nonetheless inevitable for the range-corrected XC functionals, and it is a price one has to pay to achieve a balanced description of the valence, charge-transfer, and Rydberg states at the TDDFT level of theory. Despite the overestimated energies, the CAM-B3LYP functional yielded the overall picture of the AA_2_ excited states (the nature of excited states, the energy gaps between the S_1_ and S_2_ states) that is consistent with that obtained using CC2. The B3LYP functional, on the other hand, yielded a qualitatively different picture, which will be shown below.

The common feature of the excited states that stems from the weak interactions of the AA moieties in the C_2h_ (ground state) geometry of the dimer is the appearance of closely lying pairs of states (see Figure 2) that correspond to linear combinations of diabatic states. The latter are either local excitations of one of the monomers (A*B or AB*) or charge transfer states (A^+^B^−^ or A^−^B^+^). The linear combinations of the CT states for the C_2h_ geometry represent the so-called charge-resonance (CR) states that, owing to the symmetry requirements, carry no dipole moment. The symmetric nature of the excited states can be viewed in detail in Figure 3, which displays their analysis based on the one-electron transition density matrix [59]. Note that, owing to symmetry reasons, one state of each pair is dark (having zero oscillator strength). When the dimer becomes distorted towards the geometry of the S_1_ state, in which the excitation is strongly localized on one of the AA monomers, the quasi-degenerate pairs of states gradually drift apart (in energy scale) from one another, often interacting with states from other pairs. Also, geometry distortion brings about the oscillator strength transfer between the states, so that in the S_1_ geometry both states in each pair may have comparable intensities (cf. Table 2). Note that for the planar conformation of the AA dimer, the electronic states fall into two irreducible representations (a′ and a″) of the C_s_ symmetry group. States that have different symmetries do not interact, and their potential energy surfaces (PES) may freely cross. For adiabatic states of the same symmetry, however, we observe avoided crossings, through which the natures of the states may interchange. One example of such a behavior occurs for the S_2_ (2a′) and S_3_ (3a′) states, as described by the B3LYP functional. In the B3LYP landscape of the excited states, at the C_2h_ geometry, the S_3_ (3a′) and S_4_ (4a′) states are located less than 0.2 eV above the lowest-energy pair of states. As can be seen in Figure 3, they are the charge-resonance states, which evolve into the regular charge-transfer states as the dimer loses its center of inversion (the complete account of the gradual changes in the nature of the excited states (for both TDDFT and CC2) can be seen in Appendix A of the electronic supplement). At the same time the S_3_ (3a′) state approaches its lower-energy neighbor, and their interaction is reflected in the inflections of the energy profiles at about 40% of the distortion from C_2h_ towards the S_1_ geometry. At the S_1_ geometry, the S_2_ (2a′) and S_3_ (3a′) states are completely interchanged, with the former now being of CT character. This interchange of states is also reflected by the fact that the sizeable oscillator strength in the S_1_ geometry is predicted for the 1a′ and 3a′ states (cf. Table 2), whereas the 2a′ state is much weaker, which is characteristic of the CT states for any system in which the overlap between the orbitals of the donor and the acceptor is negligible (the residual intensity is due to traces of the local excitation in the adiabatic state 2a′ state). The close proximity of the two CT states to the local excited states S_1_ (1a′) and S_2_ (2a′) is most likely, however, an artifact of B3LYP, which, just like any other GGA or hybrid exchange–correlation (XC) functional, suffers from the many-electron self-interaction error (SIE) [61,62,63,64]. The result of this error is an incorrect asymptotic of the XC potential, which decays much faster than the proper 1/r dependence. The practical outcome of the SIE is serious underestimation of the energy of CT and Rydberg states, which is the notorious problem of TDDFT.

Being free from this issue, the wave-function-based correlated CC2 method is bound to give a more balanced description of the local and CT excitations. CC2 is known to yield surprisingly good excitation energies of organic molecules, often outperforming the more complete CCSD model [65]. Our study confirms the good reputation of CC2, which locates the adiabatic energies of the S_1_ state of both the AA monomer and the dimer within the margin of a few hundredths of eV from the accurate experimental values of the supersonic jet measurements. Such close agreement is likely to be accidental (typical accuracy of CC2 for the energies of the singlet states is about 0.1–0.2 eV [65], owing to some fortuitous cancelation of errors; this excellent performance, however, has been corroborated by the CASSCF/NEVPT2 results, which are also in exceptionally good agreement with the experimental data. The charge-resonance states (and the corresponding CT ones) are located by the CC2 method slightly above 6 eV (which is indeed much higher energy with respect to the B3LYP prediction), falling between the S_7_ (5a′) and S_10_ (8a′) states. In fact, the S_7_ (5a′) state strongly mixes with the S_8_ (6a′) CT state already for the 10% distortion of the C_2h_ geometry, replacing it as S_7_ (5a′) as the geometry deviates towards the S_1_ structure by 30%. Another characteristic feature of the CC2 landscape of the excited states is a large gap of over 1.3 eV between the lowest energy pair of states and the higher lying ones. This is also in contrast to the B3LYP results, where the group of four low-energy states is energetically separated, and the gap there is considerably less than 1 eV.

The results of the CAM-B3LYP calculations are much closer to the CC2 predictions than the B3LYP ones, even if small differences in the calculated state energies lead to some qualitative discrepancies. In particular, the energies of the 1a″ and 2a″ states, being higher by some 0.3 eV in the CAM-B3LYP picture, result in their crossing with the 5a′ state that occurs for a small (10–20%) geometry deformation. In the CC2 picture, the related crossings (of the 1a″ and 3a′ states and of the 2a″ and the 5a′ states) take place for large structural deformation (80–90%). On the other hand, CAM-B3LYP locates the pair of charge-resonance states (5a′ and 6a′) about 0.2 eV lower in energy than CC2 does. As a result, the former method predicts an avoided crossing of the 4a′ (local excitation) and 5a′ (CT) states at about 80% of geometry deformation (resulting in the interchange of the states’ character, cf. Figure 3), whereas for the latter such an avoided crossing takes place between the 5a′ (local excitation) and 6a′ (CT) states and for much smaller geometry distortion (10–20%). In spite of these differences, however, the overall similarity of the CAM-B3LYP predictions and the CC2 picture is quite evident. This comes as little surprise, as the range-corrected XC functionals have been designed to obtain a more accurate description of the CT and Rydberg states by allowing the contribution of the non-local (Hartree–Fock type) exchange to gradually increase for large inter-electron distances. Improvement for the charge-separated states, however, comes at the cost of the overestimated energies for the local excitations (which is the case also for the AA monomer and dimer). Nonetheless, CAM-B3LYP yielded a much better overall picture of the excited states than the B3LYP functional, still often regarded as the first, natural choice when it comes to DFT or TDDFT calculations. While it performs quite well for the ground states of medium-sized organic molecules, its inaccuracy for the charge-separated states [66,67,68] (and also for systems with extended regions of electronic delocalization [69,70]) should be recognized and taken into account.

Note that the planar conformation of the AA dimer is an approximation, even if it is predicted by DFT. High-level coupled cluster calculations, however, revealed that in the ground state, the equilibrium geometry of the NH_2_ groups is pyramidal with an extremely low barrier of inversion of about 45 cm^−1^ [71]. The top of the barrier lies in fact well below the ground state energy level of the NH_2_ inversion, owing to which the corresponding wavefunction has its maximum for the planar conformation of the ammino group. The presence of the barrier makes the wavefunction somewhat broader than the regular gaussian that describes harmonic oscillations in the ground state. The AA can thus be regarded as dynamically planar, even though the non-planar equilibrium conformation and the inversion barrier cause subtle effects that could be detected in the supersonic jet measurements of deuterated AA [72,73]. Since changes in the state energies introduced by the assumed planarity are tiny anyway, we decided to use this approximation for the sake of clarity and to facilitate direct comparison of the excited states computed with CC2 and TDDFT (B3LYP yielded a nearly planar conformation, whereas CAM-B3LYP predicted AA to be completely planar). The picture of the excitations of the non-planar AA dimer from the CC2 calculations can be readily viewed in the electronic supplement (Appendix A).

### 3.2. Symmetry Breaking of the Electronic Structure in the S_1_ (π→π*) Excites State of AA_2_: Crucial Role of Small Geometry Distortions

In the S_1_ excited state of aromatic acid dimers, the electronic excitation is almost entirely localized in one of the monomers, BA_2_ [1,3,15], SA_2_ [20,21], and AA_2_ [22,23], whose relaxed geometry closely resembles that of the isolated AA molecule in its S_1_ state. At the same time, the geometry of the other monomer remains virtually the same as it is in the ground state. The structure of the unexcited dimers has the C_2h_ symmetry, with both monomers equivalent due to the presence of the center of inversion. Consequently, the excited states calculated for the ground state geometry are spanned by equal contributions of the diabatic states (in which either of the monomers is excited or an electron is transferred from one monomer to the other). Owing to very weak interactions between the monomers, the adiabatic excited states form closely spaced pairs corresponding to the symmetric (*a_g_*) and antisymmetric (*b_u_*) combinations of the diabatic basis states. This proximity of different electronic states violates the key condition for the Born–Oppenheimer approximation to hold, which is the energetic isolation of the potential energy surfaces for the nuclei. Consequently, the electronic wavefunctions of the excited states for the C_2h_ structure are vulnerable to even small changes in geometry. In order to illustrate this dependence, we have performed the analysis based on the one-electron TDM [74,75] for the excitations to the low-energy states of the AA dimer for a series of snapshot geometries taken while the molecule is gradually transformed from the ground state structure to the geometry optimized for the S_1_ state. Figure 3 displays the results of such analyses for the centrosymmetric (ground state) geometry, for the structure that has been distorted by 5% towards the S_1_ state geometry, and for the actual geometry of the S_1_ state (complete data are presented in Appendix A of the electronic supplement). It is clear that even such a small distortion of geometry from the centrosymmetric structure brings about unproportionally large changes in the compositions of the excited states. Apparently, the decisive symmetry breaking of the wavefunction is brought about by very small deviations from the C_2h_ structure (caused, for example, by totally non-symmetric vibrations even with the ground state amplitudes); thus, the produced non-symmetric wavefunction is then the driving force toward the fully localized S_1_ state. The above is consistent with results indicating the role of vibrational degrees of freedom in photoinduced symmetry-breaking phenomena [30,31,32,33].

This rapid symmetry breaking of the wavefunctions upon small distortions from the centrosymmetric structure is revealed by means of the one-electron TDM-based analysis, which allows for decomposition of the excited states into the diabatic components (the local excitations on either of the monomers and the charge transfer states). Figure 3 contains results of such decomposition for the centrosymmetric (C_2h_) ground state geometry, for the strongly asymmetric geometry optimized in the S_1_ electronic state, and for the C_2h_ geometry that has been distorted by 5% towards the S_1_ structure. The analyses for all the intermediate structures (vide supra) are displayed in Appendix A of the electronic supplement. For the ground state geometry, all the states consist of equal contributions of the local excitations on both monomers or of equal amounts of the A^+^B^−^ and A^−^B^+^ configurations. The latter form the charge-resonance (CR) states, which carry no net dipole moment (owing to symmetry requirements), yet the electron and hole are still separated in a dynamic fashion [76,77]. The CR states are ubiquitous in symmetric dimers, and they are often responsible for unexpectedly strong interactions in the corresponding excimers [78,79]. The ability to capture the CR states is one of the important advantages of the one-electron TDM-based analysis, as these states are hard to detect in another way. Difference density, for example, would not capture them, being symmetrically delocalized over the whole dimer.

There is a qualitative similarity between the predictions of CC2 and of the TDDFT approach when the CAM-B3LYP XC functional has been employed. Among the 12 lowest energy states, both methods yielded only one pair of the CR states at rather high energies (6.05 eV for CC2 and 5.85 eV for CAM-B3LYP). All the other states are mostly locally excited, with some small CR admixtures for 1a″, 2a″ states, and—for CC2 results only—for the 8a′ state. B3LYP functional, on the other hand, due to its tendency to underestimate the energies of states with spatially separated charges, predicts four states (among the considered twelve) to have the CR character and six more to have admixtures of this kind. Moreover, as has already been mentioned, the lowest energy pair of CR states lies just above the two lowest energy local excitations, and one of the corresponding CT states strongly mixes with the locally excited state when the geometry of the dimer relaxes towards the equilibrium structure corresponding to the S_1_ state. The change in compositions of all the electronic states is already substantial at 5% geometry distortion of the centrosymmetric structure towards that of the S_1_ state. The non-CR states become localized on either of the monomers, whereas the CR states turn into proper CT states. These changes are accompanied by increasing energy gaps between the states that are quasi-degenerate in the C_2h_ symmetry. The energy splitting between the CT states reaches 0.4–0.45 eV for the S_1_ geometry of the dimer. Interestingly, energy lowering always concerns the CT state in which the excess electron resides on the monomer with distorted geometry (by the localized S_1_ excitation). The energy of the CT state in which the electron has been transferred in the opposite direction is increased by geometry relaxation. Apparently, changing the monomer geometry from the ground state one to that of the S_1_ state stabilizes the excess negative charge and destabilizes the hole.

The differential density (DD) maps (Figure 4) corroborate the observations based on the one-electron TDM analysis. The major changes occur at initial stages of geometry deformation towards the distorted structure of the S_1_ state. It is virtually the same for 20% of the total geometry change as for the final excited-state geometry (for the complete set of DD maps for all intermediate structures, see Appendix A of the electronic supplement). Even for 10% of the geometry change, there is already distinct localization of the DD on the excited monomer (Figure 4), which reflects prominent symmetry breaking of the electronic wavefunction at the early stage of geometry relaxation.

Figure 4 also shows that the most significant decrease in electron density occurs at the nitrogen atom, while a smaller decrease can be observed at carbon atoms C1, C3, and C5 (for the numbering, please refer to Figure 5 and Figure 6). In contrast, the biggest increase in electron density is predicted for the whole carboxylic group, whereas a smaller increase occurs at the remaining carbon atoms in the ring. These dominant changes in electron density are partially compensated for by displacements of the σ electrons due to σ-π correlation. The overall pattern of excitation-induced differences in electron density is similar in both TDDFT and CC2 calculations. The most significant discrepancies concern the non-excited monomer, for which TDDFT predicts only residual changes in electron density, while the CC2 method shows a noticeable increase in C7, C2, C4, and C6, accompanied by a small decrease in density on the oxygen atoms.

The difference densities yielded by the two XC functionals we employed in our study are virtually identical for the centrosymmetric C_2h_ geometry (0% of deformation towards the S_1_ geometry) as well as for the relaxed S_1_ structure (100% deformation). Some discrepancies can be observed, however, for the intermediate structures (10% or 20% of deformation), where the difference densities on the “unexcited monomer” are noticeably larger in the B3LYP picture. CAM-B3LYP localizes the S_1_ state density faster (with respect to the geometry changes) than B3LYP does, which can be rationalized in terms of the higher contribution of the non-local (Hartree–Fock-like) exchange in the former functional. The HF method has the tendency to overlocalize the electrons, and so do the functionals that contain large non-local exchange admixtures. This unfavorable tendency can be compensated for by the non-local contributions to the correlation part, hence the success of the so-called double-hybrid functionals [80,81,82,83]. This increase in accuracy comes, however, at the price of increased computational cost, which rises steeply with the size of the basis set as the non-local correlation part scales formally like the MP2 methods (~N^5^_,_ N being the number of all the orbitals—occupied and virtual alike).

A complementary picture of the S_0_→S_1_ transition is given by the natural transition orbitals (NTOs), which give the most compact representation of the electronic configurations that span the excited state wavefunction. The NTOs calculated at the CC2 level of theory for the ground state geometry (C_2h_), the S_1_ state geometry, and a number of intermediate structures with small (5%) to moderate (20%) distortions are presented in Appendix A of the electronic supplement. Analysis of the NTOs fully corroborates the conclusions based on the one-electron TDM and on the difference densities. Even 5% geometry distortion from the centrosymmetric structure results in the 81.2% to 17.9% advantage in favor of the NTOs localized on one of the AA moieties. Moreover, 10% geometry distortions shift this imbalance further to the 91.9/7.1 ratio, whereas 20% geometry distortions result in nearly complete (96.9%) localization of the electronic excitation on one AA moiety.

### 3.3. Bond Orders for the S_0_ and S_1_ Electronic States of AA

The NEVPT2 calculations have been performed to produce the bond orders based on the one-electron density matrix (P) from the correlated calculations (in addition to verification of the nature of the excited states). The multireference approach has the advantage over the single reference ones (e.g., CC2, and even more so over the time-dependent DFT) of treating the ground state and the excited states on equal footing, thus facilitating comparison between the bond orders for the S_1_ state and the ground state. The Mayer bond orders are related to the Mulliken population analysis, being based on P and the overlap matrix S according to the formula:$$B_{AB}^{Mayer}=\sum_{s}^{on\;A}\sum_{t}^{on\;B}{\left(PS\right)}_{st}{\left(PS\right)}_{ts}$$

They share the basis set dependence of the Mulliken approach, but comparisons between bond orders obtained using the same basis set are meaningful. The Mayer bond orders also correlate well with the bond lengths satisfying the relation proposed by Pauling and co-workers [84,85,86]:$$B_{AB}=\exp\left(\frac{-(r-r_{0})}{b}\right)$$

The Mayer bond orders have long been regarded as a useful tool for studying the main group and transition metal systems [87,88]. The Mayer bond orders (B) for the AA monomer and dimer are shown in Figure 5 and Figure 6, respectively. Bond orders for the ground electronic state and for the S_1_ state have been computed for the corresponding optimum geometry for each of the electronic states. The calculations predicted noticeable changes in bond orders due to the dimerization of the AA moieties, which is related to the formation of two strong O···HO intermolecular hydrogen bonds (B_O···HO_ = 0.14) and the imposed weakening of the covalent O–H bonds by 0.09 (B_O–H_ decreases from 0.92 in the AA monomer to 0.83 in the dimer). Furthermore, the formation of the O···HO hydrogen bonds increases the contribution of the ‘ionic’ resonant structure of the COOH groups, which in turn enhances the π-electronic coupling of the C–O (and oxygen lone pairs) with C=O bond. Therefore, after dimerization, the B_C=O_ decreases by 0.16 (from 1.75 to 1.59; Figure 5 and Figure 6), while B_C–O_ increases by 0.11 (from 1.11 to 1.22). A very small competitive influence of the newly formed O···HO bonds on the existing intramolecular NH···O hydrogen bond is discernible, leading to a small decrease in the B_NH···O_ (from 0.05 in AA to 0.04 in AA_2_) and an increase in the B_N–H_ by 0.01.

The largest excitation-induced decrease in bond orders by 0.24–0.31 concerns the C1–C2, C3–C4, and C5–C6 bonds in the aromatic ring of the excited AA molecule and have similar magnitude regardless of whether it is isolated (Figure 5) or involved in the AA dimer (Figure 6). The excitation increases the π-electronic coupling of the substituents with the aromatic ring, thus increasing the B_C–COOH_ by 0.11 and B_C–N_ by 0.06, while the C1–C2 bond turns into essentially a single bond. Additionally, the excitation brings about a decrease in the C=O bond order (by 0.11 in AA_2_ and by 0.15 in AA).

The above results indicate that in the excited state the strong π-electronic coupling extends from the carbonyl oxygen of the COOH group through the benzene ring to the amino nitrogen atom, while the C1–C2 bond is not included in the π-electronic system. Upon the excitation, both hydrogen bonds formed by the carbonyl oxygen of the excited monomer M1* become stronger and increase their bond order by 0.06 and 0.03, for the NH···O and O···H, respectively. The above modifications of bond orders cause significant changes in vibrational frequencies, especially for high-frequency N–H and O–H stretches, which were observed using IR-UV and UV-IR double resonance spectroscopy techniques [22,28]. The excitation-induced modifications of bond orders impose the changes in bond lengths and related geometry changes concerning valence angles, which results in the non-zero Frank–Condon factors and band intensities in the excitation spectra of AA [38,39] and could be further investigated for AA_2_.

## 4. Conclusions

The excited state wavefunctions in the AA_2_ homodimer rapidly localize on one of the monomers after even slight distortion of geometry from the centrosymmetric structure. It is clearly revealed by the wavefunction decomposition into a pseudo-diabatic basis using one-electron TDM analysis for the 12 lowest energy excited states of the dimer. For the S_0_→S_1_ π-π * transition, it is corroborated by the electron difference density, which already at 10% of geometry distortion (towards the structure optimized for the S_1_ state) shows strong asymmetry in favor of the excited monomer. This may seem natural, since the excitation indeed localizes on one of the monomers (for which there is ample experimental evidence), but the question remains whether it is the wavefunction that follows the geometry change, or rather whether the change in the wavefunction is the driving force for the structural modifications. The answer, it seems, is neither. The breach of the central symmetry of the molecular structure is necessary to initiate the process of localization, but then the wavefunction (or electron density) takes over while the structural changes follow suit. Such behavior can be expected for any dimer of weakly interacting, identical monomers, for which in the centrosymmetric conformation the excited state wavefunctions form quasi-degenerate pairs of states, thus violating the conditions for the Born–Oppenheimer approximation. The states in each pair are of different symmetry—in the C_2h_ point group the π-π* transitions belong to *a_g_* and *b_u_* irreducible representations. When the center of symmetry is lost due to even a small totally non-symmetric molecular deformation, both states may fall into the same irreducible representation of a lower-symmetry point group. States of the same symmetry are allowed to interact and, in consequence, mix with one another. The interaction can be strong, owing to their close energetic proximity, so small geometry distortions (e.g., zero-point oscillations) may result in large changes in the nature of the interacting states. Hence, the described instability and nearly spontaneous symmetry breaking of the excited state wavefunctions happen.

The computed adiabatic energies of the S_1_ state of the AA dimer compare very favorably with the accurate (supersonic jet expansion measurements) experimental values of 3.502 eV [22,23], ranging from 3.493 eV (3.638 without the ΔZPE corrections) for CC2 to though 3.512 eV (3.629 eV) for B3LYP to 3.614 eV (3.759 eV) for NEVPT2. The only outlier was the CAM-B3LYP prediction, which located the S_1_ state at 3.837 eV (3.932 eV). On the other hand, the overall energetic landscape of excited states obtained using the CAM-B3LYP functional is qualitatively similar to that yielded by CC2. Also, the number and energetic position of the charge resonance states (or charge transfer states for non-centrosymmetric geometries) closely follow the CC2 predictions. This demonstrates the positive impact of the flexible contribution of the non-local exchange in the functional on the calculated excited states. On the other hand, the B3LYP functional (containing the fixed, 21% admixture of the non-local exchange) performs rather poorly, but for the lowest excitation of the dimer, owing to the incorrect asymptotic of the corresponding XC potential. The most striking symptom of its problems are two charge-transfer CT states of erroneously low energies that enter as intruders into the energy range occupied by the locally excited S_1_ and S_2_ states. For the electron difference densities, the overall pattern of excitation-induced changes in electron density yielded by TDDFT and CC2 is similar. The noticeable discrepancies concern the non-excited monomer, for which TDDFT predicted only residual changes in electron density, while the CC2 method yielded a sizeable increase in electron density on the carboxylic carbon atom and several carbon atoms in the aromatic ring that appear as a response to the electronic excitation (and geometry relaxation) of the other monomer.

The NEVPT2 results show noticeable changes in bond orders due to the dimerization of the AA moieties, related to the formation of two strong O···HO intermolecular hydrogen bonds that influence the carboxylic groups, increasing the contribution of the ‘ionic’ resonant structure and enhancing the π-electronic coupling within the COOH group.

In the excited monomer, the largest excitation-induced decrease in bond orders concerns every second bond in the aromatic ring (beginning from the C1–C2 one, which becomes essentially a single bond). The excitation increases the π-electronic coupling of the substituents with the aromatic ring and leads to a decrease in the C=O bond order. Thus, in the exited AA molecule, the strongly coupled π system extends from the carbonyl oxygen along the rim of the benzene ring (passing around the C1–C2 bond) to the amino N atom. In the dimer, we observe a nearly identical pattern of the excitation-induced bond order changes concerning only one of the AA moieties, while the other one remains virtually unaffected, which again demonstrates nearly complete localization of the electronic excitation. At the same time, both hydrogen bonds formed by the carbonyl oxygen atom of the excited monomer M1* become stronger and increase their fractional bond orders. The driving force of the changes in the H-bond strength upon the excitation is the increased charge density on the carboxylic group of the excited monomer.

The excitation-induced modifications of the electronic structure of the AA molecule and its dimer (symmetry breaking) result in significant changes in both the structural parameters and vibrational frequencies. These were extensively studied both experimentally and theoretically for the AA molecule and will now be further investigated for the anthranilic acid dimer.

## Figures and Tables

**Figure 1 molecules-29-05562-f001:**
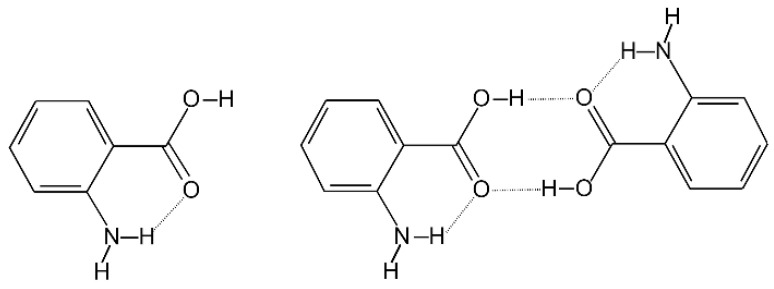
The only detected rotamer R1 of anthranilic acid (**left**) and the related rotamer of the dimer: R1:::R1 (**right**).

**Figure 2 molecules-29-05562-f002:**
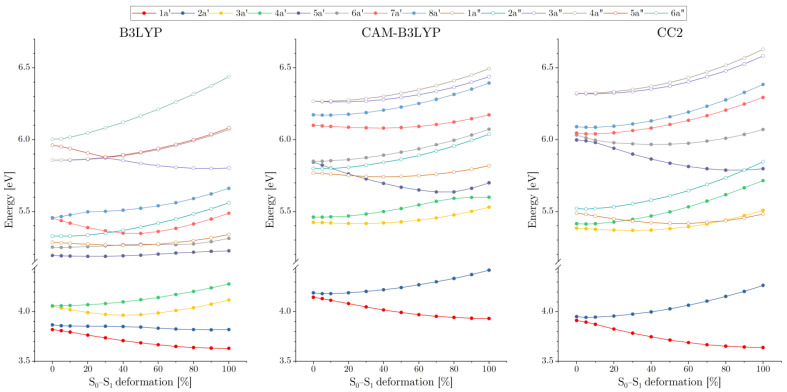
The energies of the lowest excited states of the AA dimer (with respect to the ground state energy of the C_2h_ structure) calculated at different levels of theory for a series of geometries, starting from the planar, centrosymmetric structure of the electronic ground state (0% of deformation) and gradually distorted towards the structure corresponding to the S_1_ excited state (100% of deformation).

**Figure 3 molecules-29-05562-f003:**
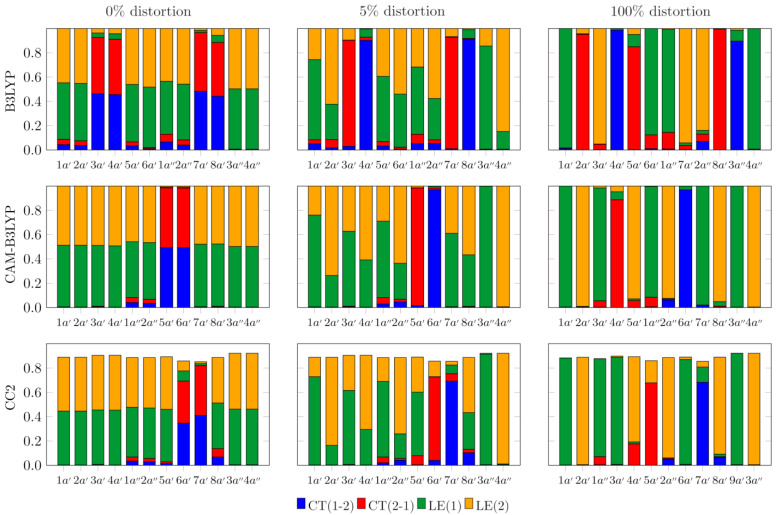
Compositions of the excited states of the AA homodimer in terms of the diabatic components (see text) based on one-electron TDM analysis calculated at different levels of theory. (1–2) in the description of the CT component means that an electron has been transferred from monomer 1 to monomer 2. For consistency of the notation, the excited states for the ground state geometry (0% if deformation) are also described within the framework of the C_s_ symmetry group.

**Figure 4 molecules-29-05562-f004:**
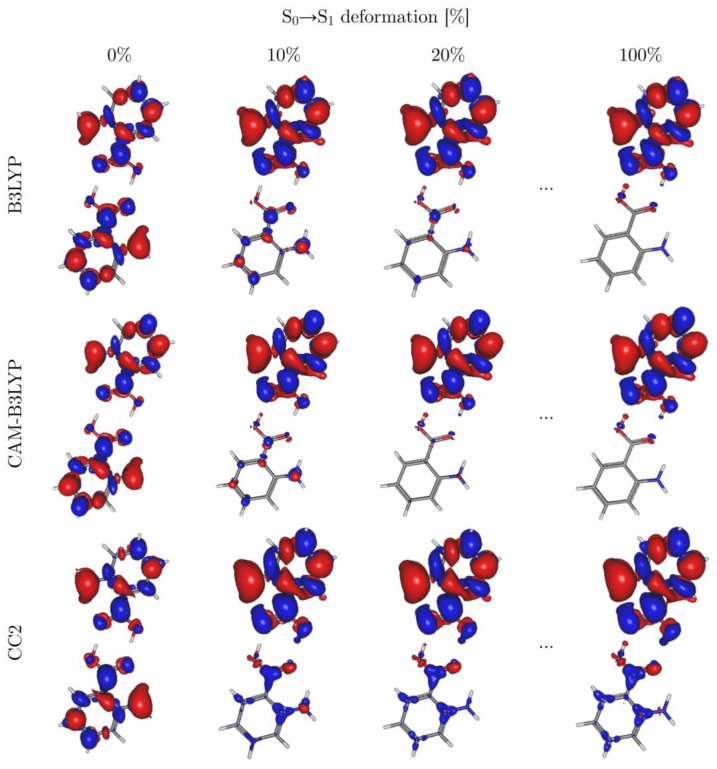
Electron difference densities calculated using CC2 and one of the variants of TDDFT for the S_0_→S_1_ excitation calculated at different degrees of geometry deformation from the centrosymmetric structure of the ground state towards the distorted geometry deformation from the centrosymmetric structure of the ground state towards the distorted geometry of the S_1_ state of the AA dimer. Blue color signifies regions with increased electron density; red color signifies regions with decreased electron density.

**Figure 5 molecules-29-05562-f005:**
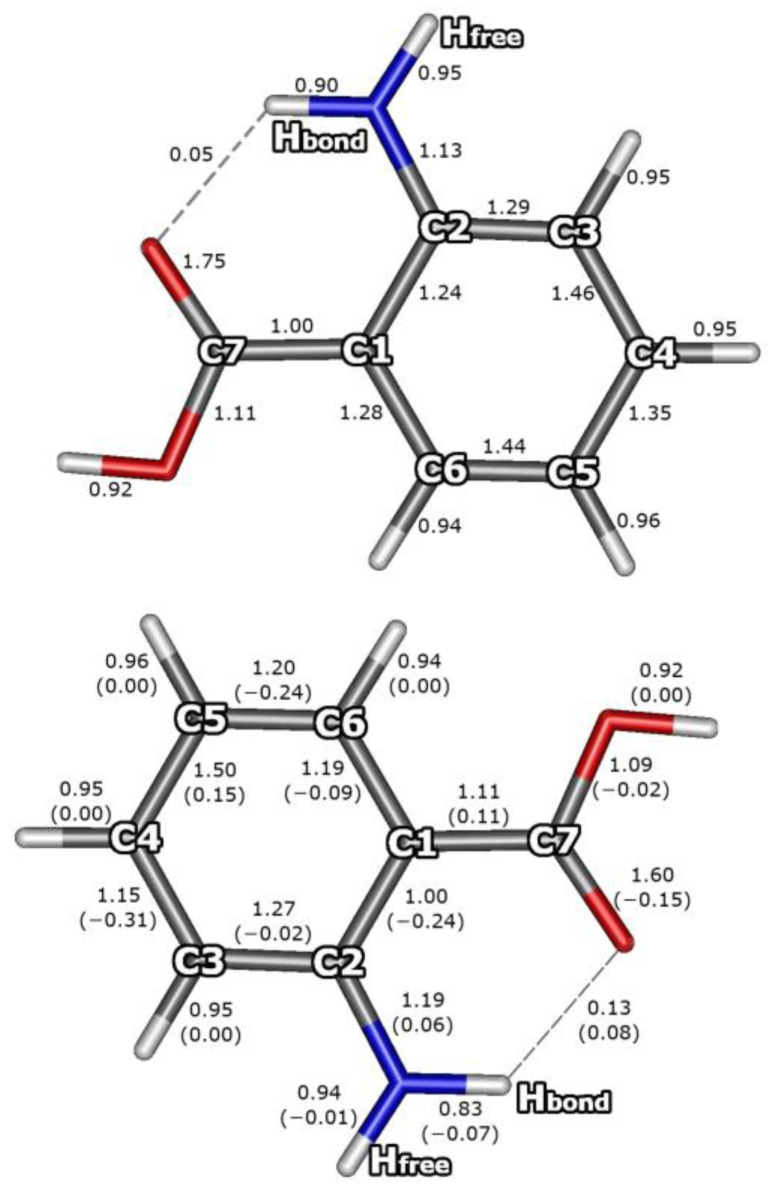
Mayer bond orders of AA monomer for the ground state (**upper panel**) and for the S_1_ state (**lower panel**). Changes in the bond orders due to the electronic excitation are given in parentheses.

**Figure 6 molecules-29-05562-f006:**
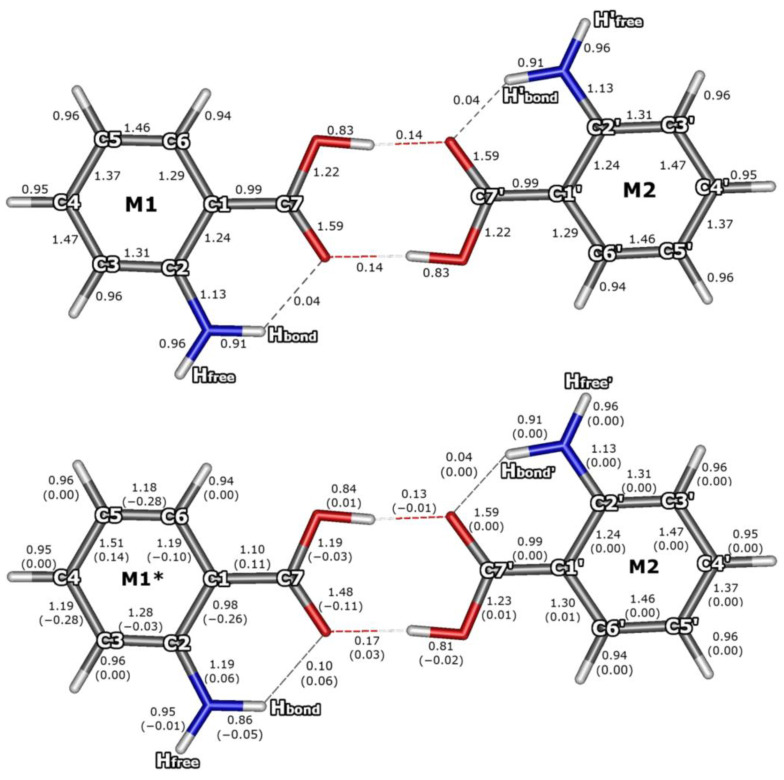
Mayer bond orders of AA dimer for the ground state (**upper panel**) and for the S_1_ state (**lower panel**). Changes in the bond orders due to the electronic excitation are given in parentheses.

**Table 1 molecules-29-05562-t001:** Vertical and adiabatic transition energies [in eV] for the S_1_ state of the AA monomer and the AA dimer, calculated at different levels of theory (see text), set against the experimental data from supersonic jet measurements. Values in parentheses are ZPE-corrected, i.e., have been corrected by differences between the zero-point energies for the S_1_ and S_0_ states.

	Monomer		Dimer	
	$S_{1}^{vert}$	$S_{1}^{ad}$	$S_{1}^{vert}$	$S_{1}^{ad}$
**B3LYP**	3.890	3.670 (3.561)	3.820	3.629 (3.512)
**CAM-B3LYP**	4.198	3.966 (3.877)	4.146	3.932 (3.837)
**CC2**	3.972	3.684 (3.552)	3.912	3.638 (3.493)
**CC2 ^a^**	4.041	3.695 (3.536)	3.986	3.649 (3.489)
**CASSCF/NEVPT2 ^b^**	3.999	3.787 (3.655)	4.000	3.759 (3.614)
**Experimental**		**3.545** [28,36,37]		**3.502** [22,23]

^a^ The values computed for the non-planar conformation of the molecules. ^b^ The zero-point vibrational energy correction taken from the CC2 calculations.

**Table 2 molecules-29-05562-t002:** Oscillator strengths for the S_0_→S_n_ electronic transitions of the a′ symmetry, calculated at different levels of theory for AA_2_ at its ground state (C_2h_) geometry and at the geometry of the S_1_ state.

	B3LYP		CAM-B3LYP		CC2	
	S_0_	S_1_	S_0_	S_1_	S_0_	S_1_
**1a′**	0.235	0.115	0.296	0.150	0.278	0.138
**2a′**	0.000 *	0.007 *	0.000	0.137	0.000	0.133
**3a′**	0.000 *	0.107	0.085	0.071	0.121	0.074
**4a′**	0.011	0.001 *	0.000	0.000 *	0.000	0.044
**5a′**	0.153	0.054	0.000 *	0.032	1.395	0.012 *
**6a′**	0.000	0.080	0.001 *	0.003 *	0.000 *	0.792
**7a′**	0.000	0.046	1.281	0.837	0.011 *	0.094 *
**8a′**	0.035	0.002	0.000	0.432	0.000	0.478

* States of charge-resonance/charge-transfer character

## Data Availability

The original contributions presented in this study are included in the article/Appendix A. Further inquiries can be directed at the corresponding authors.

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
