# Peer review of "Symmetry Breaking of Electronic Structure upon the π→π* Excitation in Anthranilic Acid Homodimer"

_molecules, 2024, doi:10.3390/molecules29235562_

Round 1
Reviewer 1 Report
Comments and Suggestions for Authors
Interesting work, using state of the art methods to tackle a tricky problem in theoretical spectroscopy. The work has been performed to a high level using methods well-suited to the problem, drawing out useful and valuable insight. These are explained in detail in clear and lucid manner, accompanied by excellent figures and tables.
I am pleased to recommend publication, and have no suggestions for improvement
Author Response
We are very grateful to the reviewer for examining the submitted manuscript and for his or her favorable opinion on our work.
Reviewer 2 Report
Comments and Suggestions for Authors
In this work, the authors examine how the low-energy singlet excited states of the anthranilic acid homodimer change due to symmetry breaking when transitioning from a ground-state geometry to the S1 state geometry. Using both ab initio methods, the authors analyze wavefunction composition through transition density matrices and difference density maps. Findings show that slight asymmetrical distortions lead to substantial changes in excited-state wavefunctions, with excitation nearly localizing on one monomer, enhancing pi-electronic coupling and strengthening hydrogen bonds in the excited monomer. CC2 reveals greater excitation-induced structural changes than TDDFT, while CAM-B3LYP provides better accuracy in charge-transfer state energies than B3LYP, though it overestimates excitation energies.
1. page 4
“All the calculations were done using the def2-TZVPP basis set on all atoms.”
The reference for the def2-tzvppd basis set is missing: Phys. Chem. Chem. Phys. 7, 3297 (2005).
2. Is D3 correction used in both geometry optimization and TDDFT calculations? This needs to be clarified.
3. How were adiabatic excitation energies calculated? Did the authors optimize the geometry at different excited states?
4. page 6, Figure 2
As shown in this figure, when deformation percentage is high, TDDFT@CAM-B3LYP fails to capture the state crossing compared to CC2, can the authors provide more discussions on this?
5. page 6, Figure 2
It is helpful to tabulate oscillator strengths of the excited states in this figure.
6. page 10, Figure 4
In addition to the density difference, natural transition orbitals can be more straightforward to visualize the excited states.
7. page 11, equation 1
Typo, “on B” should be placed on top of the summation.
Author Response
The whole response to the reviewer's remarks is contained in the included file

Reviewer 3 Report
Comments and Suggestions for Authors
In this paper, the authors aim to characterize the nature of the low energy singlet excited states of the AA2 homodimer, and their changes caused by deformation, using ab initio methods and DFT/TDDFT calculations, considering two different exchange-correlation functionals.
The work's introduction is sufficiently clear, with a good number of references related to the topic, going from carboxylic acid dimers in general, to studies concerning benzoic acid dimer in particular. All specialized terms and abbreviations are correctly defined as well.
The methods used for this research are appropriate, and I think they're robust, as the authors combine different approaches, using software like Turbomole, ORCA and TheoDORE to enhance the results obtained using CC2. This is particularly relevant when studying excited states of a system. The comparison of results obtained with different exchange-correlations functionals adds to the rigour and quality of this work. The figures and tables included in this paper are also adequate and clearly explained.
Results also acknowledge the limitations of the different approaches used in this paper, and the conclusions are quite complete and abundant, and I consider they are clearly supported by the results. The length of the paper is justified by its overall high quality.
Considering all the above reasons, I recommend this paper to be accepted in its present form. Congratulations to the authors, it's a robust, well written work.
Author Response
We would like to express our gratitude to the reviewer for careful examination of the submitted article and for all the favorable comments concerning our work. Thank you!